# CoAtNet: Marrying Convolution and Attention for All Data Sizes

**Zihang Dai, Hanxiao Liu, Quoc V. Le, Mingxing Tan**
Google Research, Brain Team
{zihangd,hanxiaol,qvl,tanmingxing}@google.com

## Abstract

Transformers have attracted increasing interests in computer vision, but they still fall behind state-of-the-art convolutional networks. In this work, we show that while Transformers tend to have larger model capacity, their generalization can be worse than convolutional networks due to the lack of the right inductive bias. To effectively combine the strengths from both architectures, we present CoAtNets (pronounced "coat" nets), a family of hybrid models built from two key insights: (1) depthwise **Co**nvolution and self-**At**tention can be naturally unified via simple relative attention; (2) vertically stacking convolution layers and attention layers in a principled way is surprisingly effective in improving generalization, capacity and efficiency. Experiments show that our CoAtNets achieve state-of-the-art performance under different resource constraints across various datasets: Without extra data, CoAtNet achieves 86.0% ImageNet top-1 accuracy; When pre-trained with 13M images from ImageNet-21K, our CoAtNet achieves 88.56% top-1 accuracy, matching ViT-huge pre-trained with 300M images from JFT-300M while using 23x less data; Notably, when we further scale up CoAtNet with JFT-3B, it achieves 90.88% top-1 accuracy on ImageNet, establishing a new state-of-the-art result.

## 1 Introduction

Since the breakthrough of AlexNet [1], Convolutional Neural Networks (ConvNets) have been the dominating model architecture for computer vision [2, 3, 4, 5]. Meanwhile, with the success of self-attention models like Transformers [6] in natural language processing [7, 8], many previous works have attempted to bring in the power of attention into computer vision [9, 10, 11, 12]. More recently, Vision Transformer (ViT) [13] has shown that with almost[1] only vanilla Transformer layers, one could obtain reasonable performance on ImageNet-1K [14] alone. More importantly, when pre-trained on large-scale weakly labeled JFT-300M dataset [15], ViT achieves comparable results to state-of-the-art (SOTA) ConvNets, indicating that Transformer models potentially have higher capacity at scale than ConvNets.

While ViT has shown impressive results with enormous JFT 300M training images, its performance still falls behind ConvNets in the low data regime. For example, without extra JFT-300M pre-training, the ImageNet accuracy of ViT is still significantly lower than ConvNets with comparable model size [5] (see Table 13). Subsequent works use special regularization and stronger data augmentation to improve the vanilla ViT [16, 17, 18], yet none of these ViT variants could outperform the SOTA *convolution-only* models on ImageNet classification given the same amount of data and computation [19, 20]. This suggests that vanilla Transformer layers may lack certain desirable inductive biases possessed by ConvNets, and thus require significant amount of data and computational resource to compensate. Not surprisingly, many recent works have been trying to incorporate the inductive biases of ConvNets into Transformer models, by imposing local receptive fields for attention

---

[1]The initial projection stage can be seen as an aggressive down-sampling convolutional stem.

35th Conference on Neural Information Processing Systems (NeurIPS 2021).

layers [21, 22] or augmenting the attention and FFN layers with implicit or explicit convolutional operations [23, 24, 25]. However, these approaches are either ad-hoc or focused on injecting a particular property, lacking a systematic understanding of the respective roles of convolution and attention when combined.

In this work, we systematically study the problem of hybridizing convolution and attention from two fundamental aspects in machine learning – generalization and model capacity. Our study shows that convolutional layers tend to have better generalization with faster converging speed thanks to their strong prior of inductive bias, while attention layers have higher model capacity that can benefit from larger datasets. Combining convolutional and attention layers can achieve better generalization and capacity; however, a key challenge here is how to effectively combine them to achieve better trade-offs between accuracy and efficiency. In this paper, we investigate two key insights: First, we observe that the commonly used depthwise convolution can be effectively merged into attention layers with simple relative attention; Second, simply stacking convolutional and attention layers, in a proper way, could be surprisingly effective to achieve better generalization and capacity. Based on these insights, we propose a simple yet effective network architecture named CoAtNet, which enjoys the strengths from both ConvNets and Transformers.

Our CoAtNet achieves SOTA performances under comparable resource constraints across different data sizes. Specifically, under the low-data regime, CoAtNet inherits the great generalization property of ConvNets thanks to the favorable inductive biases. Moreover, given abundant data, CoAtNet not only enjoys the superior scalability of Transformer models, but also achieves faster convergence and thus improved efficiency. When only ImageNet-1K is used for training, CoAtNet achieves 86.0% top-1 accuracy, matching the prior art NFNet [20] under similar computation resource and training conditions. Further, when pre-trained on ImageNet-21K with about 10M images, CoAtNet reaches 88.56% top-1 accuracy when finetuned on ImageNet-1K, matching the ViT-Huge pre-trained on JFT-300M, a 23× larger dataset. Finally, when JFT-3B is used for pre-training, CoAtNet exhibits better efficiency compared to ViT, and pushes the ImageNet-1K top-1 accuracy to 90.88% while using 1.5x less computation of the prior art set by ViT-G/14 [26].

## 2   Model

In the section, we focus on the question of how to "optimally" combine the convolution and transformer. Roughly speaking, we decompose the question into two parts:

1. How to combine the convolution and self-attention within one basic computational block?
2. How to vertically stack different types of computational blocks together to form a complete network?

The rationale of the decomposition will become clearer as we gradually reveal our design choices.

### 2.1   Merging Convolution and Self-Attention

For convolution, we mainly focus on the MBConv block [27] which employs depthwise convolution [28] to capture the spatial interaction. A key reason of this choice is that both the FFN module in Transformer and MBConv employ the design of "inverted bottleneck", which first expands the channel size of the input by 4x and later project the the 4x-wide hidden state back to the original channel size to enable residual connection.

Besides the similarity of inverted bottleneck, we also notice that both depthwise convolution and self-attention can be expressed as a per-dimension weighted sum of values in a pre-defined receptive field. Specifically, convolution relies on a fixed kernel to gather information from a local receptive field

$$y_i = \sum_{j \in \mathcal{L}(i)} w_{i-j} \odot x_j \quad \text{(depthwise convolution)}, \tag{1}$$

where $x_i, y_i \in \mathbb{R}^D$ are the input and output at position $i$ respectively, and $\mathcal{L}(i)$ denotes a local neighborhood of $i$, e.g., a 3x3 grid centered at $i$ in image processing.

In comparison, self-attention allows the receptive field to be the entire spatial locations and computes the weights based on the re-normalized pairwise similarity between the pair $(x_i, x_j)$:[2]

$$y_i = \sum_{j \in \mathcal{G}} \underbrace{\frac{\exp\left(x_i^\top x_j\right)}{\sum_{k \in \mathcal{G}} \exp\left(x_i^\top x_k\right)}}_{A_{i,j}} x_j \quad \text{(self-attention)}, \tag{2}$$

where $\mathcal{G}$ indicates the global spatial space. Before getting into the question of how to best combine them, it is worthwhile to compare their relative strengths and weaknesses, which helps to figure out the good properties we hope to retain.

- First of all, the depthwise convolution kernel $w_{i-j}$ is an input-independent parameter of static value, while the attention weight $A_{i,j}$ dynamically depends on the representation of the input. Hence, it is much easier for the self-attention to capture complicated relational interactions between different spatial positions, a property that we desire most when processing high-level concepts. However, the flexibility comes with a risk of easier overfitting, especially when data is limited.

- Secondly, notice that given any position pair $(i, j)$, the corresponding convolution weight $w_{i-j}$ only cares about the relative shift between them, i.e. $i - j$, rather than the specific values of $i$ or $j$. This property is often referred to translation equivalence, which has been found to improve generalization under datasets of limited size [29]. Due to the usage of absolution positional embeddings, standard Transformer (ViT) lacks this property. This partially explains why ConvNets are usually better than Transformers when the dataset is not enormously large.

- Finally, the size of the receptive field is one of the most crucial differences between self-attention and convolution. Generally speaking, a larger receptive field provides more contextual information, which could lead to higher model capacity. Hence, the global receptive field has been a key motivation to employ self-attention in vision. However, a large receptive field requires significantly more computation. In the case of global attention, the complexity is quadratic w.r.t. spatial size, which has been a fundamental trade-off in applying self-attention models.

Table 1: Desirable properties found in convolution or self-attention.

| Properties | Convolution | Self-Attention |
|---|:---:|:---:|
| Translation Equivariance | ✓ | |
| Input-adaptive Weighting | | ✓ |
| Global Receptive Field | | ✓ |

Given the comparison above, an ideal model should be able to combine the 3 desirable properties in Table 1. With the similar form of depthwise convolution in Eqn. (1) and self-attention in Eqn. (2), a straightforward idea that could achieve this is simply to sum a *global* static convolution kernel with the adaptive attention matrix, either after or before the Softmax normalization, i.e.,

$$y_i^{\text{post}} = \sum_{j \in \mathcal{G}} \left( \frac{\exp\left(x_i^\top x_j\right)}{\sum_{k \in \mathcal{G}} \exp\left(x_i^\top x_k\right)} + w_{i-j} \right) x_j \quad \text{or} \quad y_i^{\text{pre}} = \sum_{j \in \mathcal{G}} \frac{\exp\left(x_i^\top x_j + w_{i-j}\right)}{\sum_{k \in \mathcal{G}} \exp\left(x_i^\top x_k + w_{i-k}\right)} x_j. \tag{3}$$

Interestingly, while the idea seems overly simplified, the pre-normalization version $y^{\text{pre}}$ corresponds to a particular variant of relative self-attention [30, 31]. In this case, the attention weight $A_{i,j}$ is decided jointly by the $w_{i-j}$ of translation equivariance and the input-adaptive $x_i^\top x_j$, which can enjoy both effects depending on their relative magnitudes. Importantly, note that in order to enable the global convolution kernel without blowing up the number of parameters, we have reloaded the notation of $w_{i-j}$ as a scalar (i.e., $w \in \mathbb{R}^{O(|\mathcal{G}|)}$) rather than a vector in Eqn. (1). Another advantage of the scalar formulation of $w$ is that retrieving $w_{i-j}$ for all $(i, j)$ is clearly subsumed by computing the pairwise dot-product attention, hence resulting in minimum additional cost (see Appendix A.1). Given the benefits, we will use the Transformer block with the *pre-normalization* relative attention variant in Eqn. (3) as the key component of the proposed CoAtNet model.

---

[2]To simplify the presentation, we deliberately omit the multi-head query, key and value projections for now. In the actual implementation, we always use the multi-head projections.

## 2.2 Vertical Layout Design

After figuring out a neat way to combine convolution and attention, we next consider how to utilize it to stack an entire network.

As we have discuss above, the global context has a quadratic complexity w.r.t. the spatial size. Hence, if we directly apply the relative attention in Eqn. (3) to the raw image input, the computation will be excessively slow due to the large number of pixels in any image of common sizes. Hence, to construct a network that is feasible in practice, we have mainly three options:

(A) Perform some down-sampling to reduce the spatial size and employ the global relative attention after the feature map reaches manageable level.

(B) Enforce local attention, which restricts the global receptive field $\mathcal{G}$ in attention to a local field $\mathcal{L}$ just like in convolution [22, 21].

(C) Replace the quadratic Softmax attention with certain linear attention variant which only has a linear complexity w.r.t. the spatial size [12, 32, 33].

We briefly experimented with option (C) without getting a reasonably good result. For option (B), we found that implementing local attention involves many non-trivial shape formatting operations that requires intensive memory access. On our accelerator of choice (TPU), such operation turns out to be extremely slow [34], which not only defeats the original purpose of speeding up global attention, but also hurts the model capacity. Hence, as some recent work has studied this variant [22, 21], we will focus on option (A) and compare our results with theirs in our empirical study (Section 4).

For option (A), the down-sampling can be achieved by either (1) a convolution stem with aggressive stride (e.g., stride 16x16) as in ViT or (2) a multi-stage network with gradual pooling as in ConvNets. With these choices, we derive a search space of 5 variants and compare them in controlled experiments.

- When the ViT Stem is used, we directly stack $L$ Transformer blocks with relative attention, which we denote as $\text{ViT}_{\text{REL}}$.

- When the multi-stage layout is used, we mimic ConvNets to construct a network of 5 stages (S0, S1, S2, S3 & S4), with spatial resolution gradually decreased from S0 to S4. At the beginning of each stage, we always reduce the spatial size by 2x and increase the number of channels (see Appendix A.1 for the detailed down-sampling implementation).

  The first stage S0 is a simple 2-layer convolutional Stem and S1 always employs MBConv blocks with squeeze-excitation (SE), as the spatial size is too large for global attention. Starting from S2 through S4, we consider either the MBConv or the Transformer block, with a constraint that convolution stages must appear before Transformer stages. The constraint is based on the prior that convolution is better at processing local patterns that are more common in early stages. This leads to 4 variants with increasingly more Transformer stages, C-C-C-C, C-C-C-T, C-C-T-T and C-T-T-T, where C and T denote Convolution and Transformer respectively.

To systematically study the design choices, we consider two fundamental aspects generalization capability and model capacity: For **generalization**, we are interested in the gap between the training loss and the evaluation accuracy. If two models have the same training loss, then the model with higher evaluation accuracy has better generalization capability, since it can generalize better to unseen evaluation dataset. Generalization capability is particularly important to data efficiency when training data size is limited. For **model capacity**, we measure the ability to fit large training datasets. When training data is abundant and overfitting is not an issue, the model with higher capacity will achieve better final performance after reasonable training steps. Note that, since simply increasing the model size can lead to higher model capacity, to perform a meaningful comparison, we make sure the model sizes of the 5 variants are comparable.

To compare the generalization and model capacity, we train different variants of hybrid models on ImageNet-1K (1.3M) and JFT (>300M) dataset for 300 and 3 epochs respectively, both without any regularization or augmentation. The training loss and evaluation accuracy on both datasets are summarized in Figure 1.

- From the ImageNet-1K results, a key observation is that, in terms of *generalization capability* (i.e., gap between train and evaluation metrics), we have

$$\text{C-C-C-C} \approx \text{C-C-C-T} \geq \text{C-C-T-T} > \text{C-T-T-T} \gg \text{ViT}_{\text{REL}}.$$

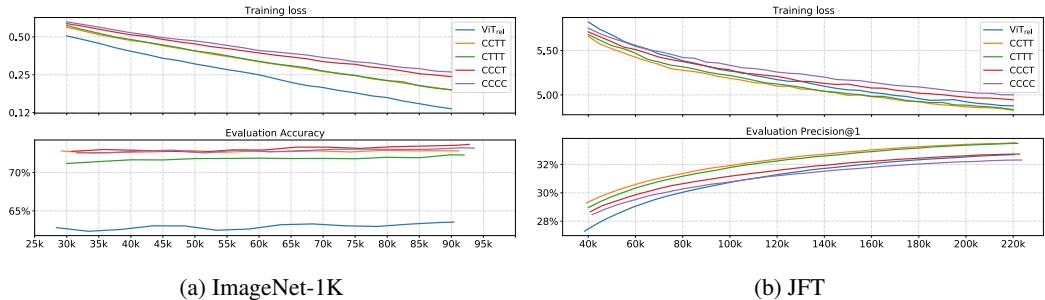

| (a) ImageNet-1K | (b) JFT |

Figure 1: Comparison for model generalization and capacity under different data size. For fair comparison, all models have similar parameter size and computational cost.

Particularly, $\text{V}_{\text{IT}_{\text{REL}}}$ is significantly worse than variants by a large margin, which we conjecture is related to the lack of proper low-level information processing in its aggressive down-sampling Stem. Among the multi-stage variants, the overall trend is that the more convolution stages the model has, the smaller the generalization gap is.

- As for *model capacity*, from the JFT comparison, both the train and evaluation metrics at the end of the training suggest the following ranking:

$$\text{C-C-T-T} \approx \text{C-T-T-T} > \text{V}_{\text{IT}_{\text{REL}}} > \text{C-C-C-T} > \text{C-C-C-C}.$$

Importantly, this suggests that simply having more Transformer blocks does NOT necessarily mean higher capacity for visual processing. On one hand, while initially worse, $\text{V}_{\text{IT}_{\text{REL}}}$ ultimately catch up with the two variants with more MBConv stages, indicating the capacity advantage of Transformer blocks. On the other hand, both C-C-T-T and C-T-T-T clearly outperforming $\text{V}_{\text{IT}_{\text{REL}}}$ suggest that the ViT stem with an aggressive stride may have lost too much information and hence limit the model capacity. More interestingly, the fact that C-C-T-T ≈ C-T-T-T indicates the for processing low-level information, static local operations like convolution could be as capable as adaptive global attention mechanism, while saving computation and memory usage substantially.

Finally, to decide between C-C-T-T and C-T-T-T, we conduct another **transferability** test[3] — we finetune the two JFT pre-trained models above on ImageNet-1K for 30 epochs and compare their transfer performances. From Table 2, it turns out that C-C-T-T achieves a clearly better transfer accuracy than C-T-T-T, despite the same pre-training performance.

Table 2: Transferability test results.

| Metric | C-C-T-T | C-T-T-T |
|---|---|---|
| Pre-training Precision@1 (JFT) | 34.40 | 34.36 |
| Transfer Accuracy 224x224 | **82.39** | 81.78 |
| Transfer Accuracy 384x384 | **84.23** | 84.02 |

Taking generalization, model capacity, transferability and efficiency into consideration, we adapt the C-C-T-T multi-stage layout for CoAtNet. More model details are included in Appendix A.1.

## 3 Related Work

**Convolutional network building blocks.** Convolutional Networks (ConvNets) have been the dominating neural architectures for many computer vision tasks. Traditionally, regular convolutions, such as ResNet blocks [3], are popular in large-scale ConvNets; in contrast, depthwise convolutions [28] are popular in mobile platforms due to its lower computational cost and smaller parameter size [27]. Recent works show that an improved inverted residual bottlenecks (MBConv [27, 35]), which is built upon depthwise convolutions, can achieve both high accuracy and better efficiency [5, 19]. As discussed in Section 2, due to the strong connection between MBConv and Transformer blocks , this paper mostly employs MBConv as convolution building blocks.

---

[3]Rigorously speaking, this test examines not only the transferability but also the generalization.

**Self-attention and Transformers.** With the key ingredients of self-attention, Transformers have been widely adopted for neural language processing and speech understanding. As an early work, stand-alone self-attention network [34] shows self-attention alone can work well for different vision tasks, though with some practical difficulties. Recently, ViT [13] applies a vanilla Transformer to ImageNet classification, and achieves impressive results after pre-training on a large-scale JFT dataset. However, ViT still largely lags behind state-of-the-art ConvNets when training data is limited. Following that, many recent works have been focused on improving vision Transformers for data efficiency and model efficiency. For a more comprehensive review of vision Transformers, we refer readers to the dedicated surveys [36, 37].

**Relative attention.** Under the general name of relative attention, there have been various variants in literature [30, 38, 39, 34, 40, 31]. Generally speaking, we can separate them into two categories: (a) the input-dependent version where the extra relative attention score is a function of the input states $f(x_i, x_j, i - j)$, and (b) the input-independent version $f(i - j)$. The variant in CoAtNet belongs to the input-independent version, and is similar to the one used in T5 [31], but unlike T5, we neither share the relative attention parameters across layers nor use the bucketing mechanism. As a benefit of the input independence, obtaining $f(i - j)$ for all $(i, j)$ pairs is computationally much cheaper than the input-dependent version on TPU. In addition, at inference time, this only needs to be computed once and cached for future use. A recent work [22] also utilizes such an input-independent parameterization, but it restricts the receptive field to a local window.

**Combining convolution and self-attention.** The idea of combining convolution and self-attention for vision recognition is not new. A common approach is to augment the ConvNet backbone with explicit self-attention or non-local modules [9, 10, 11, 12], or to replace certain convolution layers with standard self-attention [11] or a more flexible mix of linear attention and convolution [41]. While self-attention usually improves the accuracy, they often come with extra computational cost and hence are often regarded as an add-on to the ConvNets, similar to squeeze-and-excitation [42] module. In comparison, after the success of ViT and ResNet-ViT [13], another popular line of research starts with a Transformer backbone and tries to incorporate explicit convolution or some desirable properties of convolution into the Transformer backbone [25, 24, 23, 22, 21, 43, 44].

While our work also belongs to this category, we show that our relative attention instantiation is a natural mixture of depthwise convolution and content-based attention with minimum additional cost. More importantly, starting from the perspectives of generalization and model capacity, we take a systematic approach to the vertical layout design and show how and why different network stages prefer different types of layers. Therefore, compared to models that simply use an off-the-shelf ConvNet as the stem layer, such as ResNet-ViT [13], CoAtNet also scales the Convolution stage (S2) when the overall size increases. On the other hand, compared to models employing local attention [22, 21], CoAtNet consistently uses full attention for S3 & S4 to ensure the model capacity, as S3 occupies the majority of the computation and parameters.

## 4 Experiments

In this section, we compare CoAtNet with previous results under comparable settings. For completeness, all the hyper-parameters not mentioned here are included in Appendix A.2.

### 4.1 Experiment Setting

**CoAtNet model family.** To compare with existing models of different sizes, we also design a family of CoAtNet models as summarized in Table 3. Overall, we always double the number of channels from S1 to S4, while ensuring the width of the Stem S0 to be smaller or equal to that of S1. Also, for simplicity, when increasing the depth of the network, we only scale the number of blocks in S2 and S3.

**Evaluation Protocol.** Our experiments focus on image classification. To evaluate the performance of the model across different data sizes, we utilize three datasets of increasingly larger sizes, namely ImageNet-1K (1.28M images), ImageNet-21K (12.7M images) and JFT (300M images). Following previous works, we first pre-train our models on each of the three datasets at resolution 224 for 300, 90 and 14 epochs respectively. Then, we finetune the pre-trained models on ImageNet-1K at the desired

Table 3: L denotes the number of blocks and D denotes the hidden dimension (#channels). For all Conv and MBConv blocks, we always use the kernel size 3. For all Transformer blocks, we set the size of each attention head to 32, following [22]. The expansion rate for the inverted bottleneck is always 4 and the expansion (shrink) rate for the SE is always 0.25.

| Stages | Size | CoAtNet-0 | CoAtNet-1 | CoAtNet-2 | CoAtNet-3 | CoAtNet-4 |
|---|---|---|---|---|---|---|
| S0-Conv | $1/2$ | L=2 D=64 | L=2 D=64 | L=2 D=128 | L=2 D=192 | L=2 D=192 |
| S1-MbConv | $1/4$ | L=2 D=96 | L=2 D=96 | L=2 D=128 | L=2 D=192 | L=2 D=192 |
| S2-MBConv | $1/8$ | L=3 D=192 | L=6 D=192 | L=6 D=256 | L=6 D=384 | L=12 D=384 |
| S3-TFM$_{Rel}$ | $1/16$ | L=5 D=384 | L=14 D=384 | L=14 D=512 | L=14 D=768 | L=28 D=768 |
| S4-TFM$_{Rel}$ | $1/32$ | L=2 D=768 | L=2 D=768 | L=2 D=1024 | L=2 D=1536 | L=2 D=1536 |

resolutions for 30 epochs and obtain the corresponding evaluation accuracy. One exception is the ImageNet-1K performance at resolution 224, which can be directly obtained at the end of pre-training. Note that similar to other models utilizing Transformer blocks, directly evaluating models pre-trained on ImageNet-1K at a larger resolution without finetuning usually leads to performance drop. Hence, finetuning is always employed whenever input resolution changes.

**Data Augmentation & Regularization.** In this work, we only consider two widely used data augmentations, namely RandAugment [45] and MixUp [46], and three common techniques, including stochastic depth [47], label smoothing [48] and weight decay [49], to regularize the model. Intuitively, the specific hyper-parameters of the augmentation and regularization methods depend on model size and data scale, where strong regularization is usually applied for larger models and smaller dataset.

Under the general principle, a complication under the current paradigm is how to adjust the regularization for pre-training and finetuning as data size can change. Specifically, we have an interesting observation that if a certain type of augmentation is entirely disabled during pre-training, simply turning it on during fine-tuning would most likely harm the performance rather than improving. We conjecture this could be related to data distribution shift. As a result, for certain runs of the proposed model, we deliberately apply RandAugment and stochastic depth of a small degree when pre-training on the two larger datasets, ImageNet21-K and JFT. Although such regularization can harm the pre-training metrics, this allows more versatile regularization and augmentation during finetuning, leading to improved down-stream performances.

## 4.2 Main Results

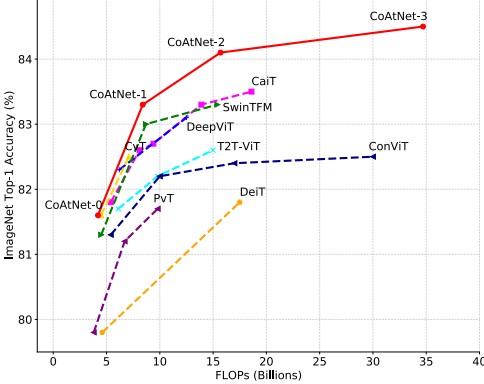

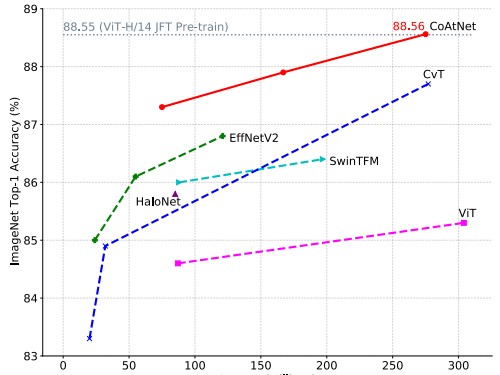

Figure 2: Accuracy-to-FLOPs scaling curve under ImageNet-1K only setting at 224x224.

Figure 3: Accuracy-to-Params scaling curve under ImageNet-21K $\Rightarrow$ ImageNet-1K setting.

**ImageNet-1K** The experiment results with only the ImageNet-1K dataset are shown in Table 4. Under similar conditions, the proposed CoAtNet models not only outperform ViT variants, but also match the best convolution-only architectures, i.e., EfficientNet-V2 and NFNets. Additionally, we also visualize the all results at resolution 224x224 in Fig. 2. As we can see, CoAtNet scales much better than previous model with attention modules.

Table 4: Model performance on ImageNet. `1K only` denotes training on ImageNet-1K only; `21K+1K` denotes pre-training on ImageNet-21K and finetuning on ImageNet-1K; `PT-RA` denotes applying RandAugment during 21K pre-training, and `E150` means 150 epochs of 21K pre-training, which is longer than the standard 90 epochs. More results are in Appendix A.3.

| | Models | Eval Size | #Params | #FLOPs | ImageNet Top-1 Accuracy | |
| --- | --- | --- | --- | --- | --- | --- |
| | | | | | 1K only | 21K+1K |
| Conv Only | EfficientNet-B7 | $600^2$ | 66M | 37B | 84.7 | - |
| | EfficientNetV2-L | $480^2$ | 121M | 53B | 85.7 | 86.8 |
| | NFNet-F3 | $416^2$ | 255M | 114.8B | 85.7 | - |
| | NFNet-F5 | $544^2$ | 377M | 289.8B | **86.0** | - |
| ViT-Stem TFM | DeiT-B | $384^2$ | 86M | 55.4B | 83.1 | - |
| | ViT-L/16 | $384^2$ | 304M | 190.7B | - | 85.3 |
| | CaiT-S-36 | $384^2$ | 68M | 48.0B | 85.0 | - |
| | DeepViT-L | $224^2$ | 55M | 12.5B | 83.1 | - |
| Multi-stage TFM | Swin-B | $384^2$ | 88M | 47.0B | 84.2 | 86.0 |
| | Swin-L | $384^2$ | 197M | 103.9B | - | 86.4 |
| Conv+TFM | BotNet-T7 | $384^2$ | 75.1M | 45.8B | 84.7 | - |
| | LambdaResNet-420 | $320^2$ | - | - | 84.8 | - |
| | T2T-ViT-24 | $224^2$ | 64.1M | 15.0B | 82.6 | - |
| | CvT-21 | $384^2$ | 32M | 24.9B | 83.3 | - |
| | CvT-W24 | $384^2$ | 277M | 193.2B | - | **87.7** |
| **Conv+TFM (ours)** | CoAtNet-0 | $224^2$ | 25M | 4.2B | 81.6 | - |
| | CoAtNet-1 | $224^2$ | 42M | 8.4B | 83.3 | - |
| | CoAtNet-2 | $224^2$ | 75M | 15.7B | 84.1 | 87.1 |
| | CoAtNet-3 | $224^2$ | 168M | 34.7B | 84.5 | 87.6 |
| | CoAtNet-0 | $384^2$ | 25M | 13.4B | 83.9 | - |
| | CoAtNet-1 | $384^2$ | 42M | 27.4B | 85.1 | - |
| | CoAtNet-2 | $384^2$ | 75M | 49.8B | 85.7 | 87.1 |
| | CoAtNet-3 | $384^2$ | 168M | 107.4B | 85.8 | 87.6 |
| | CoAtNet-4 | $384^2$ | 275M | 189.5B | - | 87.9 |
| | + PT-RA | $384^2$ | 275M | 189.5B | - | 88.3 |
| | + PT-RA-E150 | $384^2$ | 275M | 189.5B | - | 88.4 |
| | CoAtNet-2 | $512^2$ | 75M | 96.7B | 85.9 | 87.3 |
| | CoAtNet-3 | $512^2$ | 168M | 203.1B | **86.0** | 87.9 |
| | CoAtNet-4 | $512^2$ | 275M | 360.9B | - | 88.1 |
| | + PT-RA | $512^2$ | 275M | 360.9B | - | 88.4 |
| | + PT-RA-E150 | $512^2$ | 275M | 360.9B | - | **88.56** |

**ImageNet-21K**  As we can see from Table 4 and Fig. 3, when ImageNet-21K is used for pre-training, the advantage of CoAtNet becomes more obvious, substantially outperforming all previous models. Notably, the best CoAtNet variant achieves a top-1 accuracy of 88.56%, matching the ViT-H/14 performance of 88.55%, which requires pre-training the 2.3x larger ViT model on a 23x larger proprietary weakly labeled dataset (JFT) for 2.2x more steps. This marks a dramatic improvement in both data efficiency and computation efficiency.

**JFT**  Finally, in Table 5, we further evaluate CoAtNet under the large-scale data regime with JFT-300M and JFT-3B. Encouragingly, our CoAtNet-4 can almost match the best previous performance with JFT-300M set by NFNet-F4+, while being 2x more efficient in terms of both TPU training time and parameter count. When we scale up the model to consume similar training resource as NFNet-F4+, CoAtNet-5 reaches 89.77% on top-1 accuracy, outperforming previous results under comparable settings.

Moreover, as we further push the training resource towards the level used by ViT-G/14 and utilize the same JFT-3B dataset of an even larger size [26], with over 4x less computation, CoAtNet-6 is able to

Table 5: Performance Comparison on large-scale JFT dataset. `TPUv3-core-days` denotes the pre-training time, *Top-1 Accuracy* denotes the finetuned accuracy on ImageNet. Note that the last 3 rows use a larger dataset JFT-3B [26] for pre-training, while others use JFT-300M [15]. See Appendix A.2 for the size details of CoAtNet-5/6/7. [†]: Down-sampling in the MBConv block is achieved by stride-2 Depthwise Convolution. [⋄]: ViT-G/14 computation consumption is read from Fig. 1 of the paper [26].

| Models | Eval Size | #Params | #FLOPs | TPUv3-core-days | Top-1 Accuracy |
|---|---|---|---|---|---|
| ResNet + ViT-L/16 | $384^2$ | 330M | - | - | 87.12 |
| ViT-L/16 | $512^2$ | 307M | 364B | 0.68K | 87.76 |
| ViT-H/14 | $518^2$ | 632M | 1021B | 2.5K | 88.55 |
| NFNet-F4+ | $512^2$ | 527M | 367B | 1.86K | 89.2 |
| CoAtNet-3[†] | $384^2$ | 168M | 114B | 0.58K | 88.52 |
| CoAtNet-3[†] | $512^2$ | 168M | 214B | 0.58K | 88.81 |
| CoAtNet-4 | $512^2$ | 275M | 361B | 0.95K | 89.11 |
| CoAtNet-5 | $512^2$ | 688M | 812B | 1.82K | 89.77 |
| ViT-G/14 | $518^2$ | 1.84B | 5160B | >30K[⋄] | 90.45 |
| CoAtNet-6 | $512^2$ | 1.47B | 1521B | 6.6K | 90.45 |
| CoAtNet-7 | $512^2$ | 2.44B | 2586B | 20.1K | **90.88** |

match the performance of ViT-G/14 of 90.45%, and with 1.5x less computation, CoAtNet-7 achieves 89.77% on top-1 accuracy 90.88%, achieving the new state-of-the-art performance.

## 4.3 Ablation Studies

In this section, we will ablate our design choices for CoAtNet.

Firstly, we study the importance of the relative attention from combining convolution and attention into a single computation unit. Specifically, we compare two models, one with the relative attention and the other without, under both the ImageNet-1K alone and ImageNet-21K transfer setting. As we can see from Table 6, when only the ImageNet-1K is used, relative attention clearly outperforms the standard attention, indicating a better generalization. In addition, under the ImageNet-21K transfer setting, the relative attention variant achieves a substantially better transfer accuracy, despite their very close pre-training performances. This suggests the main advantage of relative attention in visual processing is not in higher capacity but in better generalization.

Table 6: Ablation on relative attention.

| Setting | Metric | With Rel-Attn | Without Rel-Attn |
|---|---|---|---|
| ImageNet-1K | Accuracy ($224^2$) | 84.1 | 83.8 |
| | Accuracy ($384^2$) | 85.7 | 85.3 |
| ImageNet-21K $\Rightarrow$ ImageNet-1K | Pre-train Precision@1 ($224^2$) | 53.0 | 52.8 |
| | Finetune Accuracy ($384^2$) | 87.9 | 87.4 |

Table 7: Ablation on architecture layout.

| Setting | Models | Layout | Top-1 Accuracy |
|---|---|---|---|
| ImageNet-1K | V0: CoAtNet-2 | [2, 2, 6, 14, 2] | 84.1 |
| | V1: S2 ⇐ S3 | [2, 2, 2, 18, 2] | 83.4 |
| | V2: S2 ⇒ S3 | [2, 2, 8, 12, 2] | 84.0 |
| ImageNet-21K $\Rightarrow$ ImageNet-1K | V0: CoAtNet-3 | [2, 2, 6, 14, 2] | 53.0 → 87.6 |
| | V1: S2 ⇐ S3 | [2, 2, 2, 18, 2] | 53.0 → 87.4 |

Secondly, as `S2` with MBConv blocks and `S3` with relative Transformer blocks occupy most of the computation of the CoAtNet, a question to ask is how to split the computation between `S2` (MBConv) and `S3` (Transformer) to achieve a good performance. In practice, it boils down to deciding the number of blocks to have in each stage, which we will refer to as "layout" design. For this purpose, we compare a few different layouts that we experimented with in Table 7.

Table 8: Ablation on head size and normalization type.

| Setting | Models | Image Size | Top-1 Accuracy |
|---|---|---|---|
| | CoAtNet-2 | $224^2$ | 84.1 |
| ImageNet-1K | Head size: $32 \rightarrow 64$ | $224^2$ | 83.9 |
| | Norm type: BN $\rightarrow$ LN | $224^2$ | 84.1 |
| ImageNet-21K | CoAtNet-3 | $384^2$ | 87.9 |
| $\Rightarrow$ ImageNet-1K | Norm type: BN $\rightarrow$ LN | $384^2$ | 87.8 |

- If we keep the total number of blocks in S2 and S3 fixed and vary the number in each stage, we observe that V0 is a sweet spot between V1 and V2. Basically, having more Transformer blocks in S3 generally leads to better performance until the number of MBConv blocks in S2 is too small to generalize well.
- To further evaluate whether the sweet spot also holds in the transfer setting, where a higher capacity is often regarded more important, we further compare V0 and V1 under the ImageNet-21K transferring to ImageNet-1K setup. Interestingly, despite that V1 and V0 have the same performance during ImageNet-21K pre-training, the transfer accuracy of V1 clearly falls behind V0. Again, this suggests the importance of convolution in achieving good transferability and generalization.

Lastly, we study two choices of model details, namely the dimension of each attention (default to 32) head as well as the type of normalization (default to `BatchNorm`) used in MBConv blocks. From Table 8, we can see increasing head size from 32 to 64 can slightly hurt performance, though it actually improves the TPU speed by a significant amount. In practice, this will be a quality-speed trade-off one can make. On the other hand, `BatchNorm` and `LayerNorm` have almost the same performance, while `BatchNorm` is 10 - 20% faster on TPU depending on the per-core batch size.

## 5 Conclusion

In this paper, we systematically study the properties of convolutions and Transformers, which leads to a principled way to combine them into a new family of models named CoAtNet. Extensive experiments show that CoAtNet enjoys both good generalization like ConvNets and superior model capacity like Transformers, achieving state-of-the-art performances under different data sizes and computation budgets.

Note that this paper currently focuses on ImageNet classification for model development. However, we believe our approach is applicable to broader applications like object detection and semantic segmentation. We will leave them for future work.

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
