# A  Appendix

## A.1  Model Details

First of all, the overview of CoAtNet is illustrated in Fig. 4.

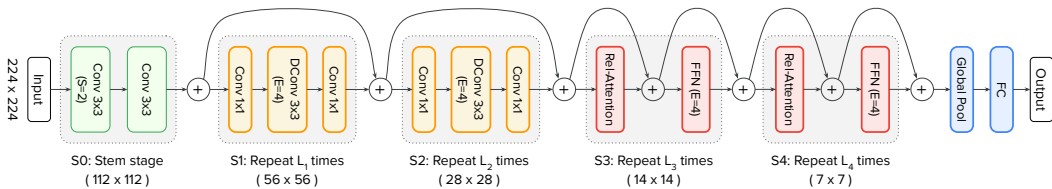

Figure 4: Overview of the proposed CoAtNet.

**2D Relative Attention**  To implement the pre-norm version of relative attention in Eqn. 3 for 2D images of size $[H \times W]$, for *each head*, we create a trainable parameter $\mathbf{P}$ of size $[(2H-1) \times (2W-1)]$, as the maximum distance is $2H-1$ and $2W-1$ respectively. Then, for two spatial locations $(i, j)$ and $(i', j')$, the corresponding relative bias is $P_{i-i'+H, j-j'+W}$ under 1-based indexing. For implementation, we need to index $H^2W^2$ elements from the $[(2H-1) \times (2W-1)]$ matrix. On TPU, we utilize two einsums, along the height and width axis respectively, to index the relative bias with complexity $O(HW(H+W))$, which is strictly subsumed by the $O(H^2W^2D)$ attention complexity. On GPUs, the indexing can be done more efficiently with gather, which only requires memory access. Note that, at inference time, indexing the $H^2W^2$ elements from the $[(2H-1) \times (2W-1)]$ matrix can be pre-computed and cached to further increase the throughput.

When finetuned on a larger resolution, we simply use bi-linear interpolation to increase the size $[(2H-1) \times (2W-1)]$ to the desired size $[(2H'-1) \times (2W'-1)]$ for any $H' > H$ and $W' > W$.

**Pre-Activation**  To promote homogeneity in the model architecture, we consistently use pre-activation structure [50] for both the MBConv and the Transformer block, i.e.,

$$\mathbf{x} \leftarrow \mathbf{x} + \texttt{Module}\left(\texttt{Norm}(\mathbf{x})\right),$$

where Module denotes the MBConv, Self-Attention or FFN module, while Norm corresponds to BatchNorm for MBConv and LayerNorm for Self-Attention and FFN. We have experimented with using LayerNorm in the MBConv block, which achieves the same performance while being significantly slower on our accelerator (TPU). In general, we recommend whichever is faster on your device. Following the same spirit, Gaussian Error Linear Units (GELUs) [51] is used as the activation function in both the MBConv blocks and Transformer blocks.

**Down-Sampling**  For the first block inside each stage from S1 to S4, down-sampling is performed independently for the residual branch and the identity branch. Specifically, for the Transformer block, the standard max pooling of stride 2 is directly applied to the input states of both branches of the self-attention module, similar to Funnel Transformer [52]. Also, a channel projection is applied to the identity branch to enlarge the hidden size. Hence, the down-sampling self-attention module can be expressed as

$$\mathbf{x} \leftarrow \texttt{Proj}(\texttt{Pool}(\mathbf{x})) + \texttt{Attention}\left(\texttt{Pool}(\texttt{Norm}(\mathbf{x}))\right). \tag{4}$$

As for the MBConv block, the down-sampling in the residual branch is instead achieved by using a stride-2 convolution to the normalized inputs, i.e.,

$$\mathbf{x} \leftarrow \texttt{Proj}(\texttt{Pool}(\mathbf{x})) + \texttt{Conv}\left(\texttt{DepthConv}\left(\texttt{Conv}\left(\texttt{Norm}(\mathbf{x}), \texttt{stride}=2\right)\right)\right). \tag{5}$$

This is different from the standard MBConv where the down-sampling is done by applying stride-2 depthwise convolution to the inverted bottleneck hidden states. We later found using stride-2 depthwise convolution is helpful but slower when model is small but not so much when model scales, as shown in Table 9. Hence, if not mentioned otherwise, numbers reported in the main text uses the down-sampling implementation in Eqn. (5). In practice, this could be yet another quality-speed trade-off one can tweak for smaller models.

Table 9: The effect of performing down-sampling in first Conv v.s. the Depthwise Conv.

| Models | Eval Size | #Params | #FLOPs | ImageNet Top-1 Accuracy |
|---|---|---|---|---|
| CoAtNet-0 | $224^2$ | 25M | 4.2B | 81.6 |
| Strided DConv | $224^2$ | 25M | 4.6B | 82.0 |
| CoAtNet-1 | $224^2$ | 42M | 8.4B | 83.3 |
| Strided DConv | $224^2$ | 42M | 8.8B | 83.5 |
| CoAtNet-2 | $224^2$ | 75M | 15.7B | 84.1 |
| Strided DConv | $224^2$ | 75M | 16.6B | 84.1 |

**Classification head** Instead of adding an additional `<cls>` token as in ViT to perform classification, we apply global average pooling to the last-stage output to get the representation for simplicity.

## A.2 Hyper-Parameters

Table 10: Hyper-parameters used in the main experiments. The slash sign " / " is used to separate the different hyper-parameters used for various CoAtNet model sizes. $^\diamond$: For finetuning the slightly larger CoAtNet-3, RandAugment of 2, 20 is used. $^\dagger$: RandAugment of 2, 5 is applied to the PT-RA variants of CoAtNet-4 in Table 14.

| Hyper-parameter | ImageNet-1K | | ImageNet-21K | | JFT | |
|---|---|---|---|---|---|---|
| | Pre-Training (CoAtNet-0/1/2/3) | Finetuning | Pre-Training (CoAtNet-2/3/4) | Finetuning | Pre-Training (CoAtNet-3/4/5) | Finetuning |
| Stochastic depth rate | 0.2 / 0.3 / 0.5 / 0.7 | | 0.3 / 0.5 / 0.7 | | 0.0 / 0.1 / 0.0 | 0.1 / 0.3 / 0.2 |
| Center crop | True | False | True | False | True | False |
| RandAugment | 2, 15 | 2, 15$^\diamond$ | None / None / 2, 5$^\dagger$ | | 2, 5 | 2, 5 |
| Mixup alpha | 0.8 | 0.8 | None | None | None | None |
| Loss type | Softmax | Softmax | Sigmoid | Softmax | Sigmoid | Softmax |
| Label smoothing | 0.1 | 0.1 | 0.0001 | 0.1 | 0.0001 | 0.1 |
| Train epochs | 300 | 30 | 90 | 30 | 14 | 30 |
| Train batch size | 4096 | 512 | 4096 | 1024 | 4096 | 512 |
| Optimizer type | AdamW | AdamW | AdamW | AdamW | AdamW | AdamW |
| Peak learning rate | 1e-3 | 5e-5 | 1e-3 | 5e-5 | 1e-3 / 5e-4 / 5e-4 | 5e-5 |
| Min learning rate | 1e-5 | 5e-5 | 1e-5 | 5e-5 | 1e-5 | 5e-5 |
| Warm-up | 10K steps | None | 5 epochs | None | 20K steps | None |
| LR decay schedule | Cosine | None | Linear | None | Linear | None |
| Weight decay rate | 0.05 | 1e-8 | 0.01 | 1e-8 | 0.01 | 1e-8 |
| Gradient clip | 1.0 | 1.0 | 1.0 | 1.0 | 1.0 | 1.0 |
| EMA decay rate | None | 0.9999 | None | 0.9999 | None | 0.9999 |

The hyper-parameters used for the main experiments presented in Section 4 are summarized in Table 10.

The model size of CoAtNet-5 used in the JFT experiment is summarized in Table 11. Different from the standard CoAtNet models in Table 3, we set the size of each attention head to 64 rather than 32 for CoAtNet-5, as this achieves a better speed-performance trade-off as discussed in Section 4.3.

Table 11: CoAtNet-5 model sizes.

| Stages | Size | CoAtNet-5 | |
|---|---|---|---|
| S0-Conv | $^1/_2$ | L=2 | D=192 |
| S1-MbConv | $^1/_4$ | L=2 | D=256 |
| S2-MBConv | $^1/_8$ | L=12 | D=512 |
| S3-TFM$_{Rel}$ | $^1/_{16}$ | L=28 | D=1280 |
| S4-TFM$_{Rel}$ | $^1/_{32}$ | L=2 | D=2048 |

For CoAtNet-6 and CoAtNet-7, to reduce the memory consumption, we move $^2/_3$ of the MBConv blocks of S2 into S3 and double its hidden dimension. While this modification does not change the

complexity in terms of FLOPs, this will reduce the activation related memory usage of these MBConv blocks by half, which enables us to build a larger model. With this adjustment, the S3 becomes a stage of mixed block types and hidden dimensions. In addition, we increase the attention head size to 128 further to boost the speed-performance trade-off. The specific sizes are summarized in Table 12. Basically, CoAtNet-6 and CoAtNet-7 share the same depth but differ in width.

Table 12: Model sizes for the scaled models.

| Stages | Size | CoAtNet-6 | | CoAtNet-7 | |
|---|---|---|---|---|---|
| S0-Conv | $1/2$ | L=2 | D=192 | L=2 | D=192 |
| S1-MbConv | $1/4$ | L=2 | D=192 | L=2 | D=256 |
| S2-MBConv | $1/8$ | L=4 | D=384 | L=4 | D=512 |
| S3-MBConv | $1/16$ | L=8 | D=768 | L=8 | D=1024 |
| S3-TFM$_{\text{Rel}}$ | | L=42 | D=1536 | L=42 | D=2048 |
| S4-TFM$_{\text{Rel}}$ | $1/32$ | L=2 | D=2048 | L=2 | D=3072 |

## A.3 Complete Comparison

Table 13: Complete comparison under the ImageNet-1K only setting.

| Models | | Eval Size | #Params | #FLOPs | Top-1 Accuracy |
|---|---|---|---|---|---|
| Conv Only | ResNet-RS-152 | $256^2$ | 87M | 31B | 83.0 |
| | ResNet-RS-420 | $320^2$ | 192M | 128B | 84.4 |
| | NFNet-F0 | $256^2$ | 72M | 12.4B | 83.6 |
| | NFNet-F1 | $320^2$ | 133M | 35.5B | 84.7 |
| | NFNet-F2 | $352^2$ | 194M | 62.6B | 85.1 |
| | NFNet-F3 | $416^2$ | 255M | 114.8B | 85.7 |
| | NFNet-F4 | $512^2$ | 316M | 215.2B | 85.9 |
| | NFNet-F5 | $544^2$ | 377M | 289.8B | **86.0** |
| | ENetV2-S | $384^2$ | 24M | 8.8B | 83.9 |
| | ENetV2-M | $480^2$ | 55M | 24B | 85.1 |
| | ENetV2-L | $480^2$ | 121M | 53B | 85.7 |
| ViT-Stem TFM Only | DeiT-S | $224^2$ | 22M | 4.6B | 79.8 |
| | DeiT-B | $224^2$ | 86M | 17.5B | 81.8 |
| | DeiT-B | $384^2$ | 86M | 55.4B | 83.1 |
| | CaiT-S-24 | $224^2$ | 46.9M | 9.4B | 82.7 |
| | CaiT-S-36 | $224^2$ | 68.2M | 13.9B | 83.3 |
| | CaiT-M-24 | $224^2$ | 185.9M | 36.0B | 83.4 |
| | CaiT-S-24 | $384^2$ | 46.9M | 32.2B | 84.3 |
| | CaiT-S-36 | $384^2$ | 68M | 48.0B | 85.0 |
| | CaiT-M-24 | $384^2$ | 185.9M | 116.1B | 84.5 |
| | DeepViT-S | $224^2$ | 27M | 6.2B | 82.3 |
| | DeepViT-L | $224^2$ | 55M | 12.5B | 83.1 |
| Multi-Stage TFM Only | PVT-Small | $224^2$ | 24.5M | 3.8B | 79.8 |
| | PVT-Medium | $224^2$ | 44.2M | 6.7B | 81.2 |
| | PVT-Large | $224^2$ | 61.5M | 9.8B | 81.7 |
| | Swin-T | $224^2$ | 29M | 4.5B | 81.3 |
| | Swin-S | $224^2$ | 50M | 8.7B | 83.0 |
| | Swin-B | $224^2$ | 88M | 15.4B | 83.3 |
| | Swin-B | $384^2$ | 88M | 47.0B | 84.2 |
| Multi-Stage Conv+TFM | BotNet-T7 | $384^2$ | 75.1M | 45.80B | 84.7 |
| | LambdaResNet-420 | $320^2$ | - | - | 84.8 |
| | T2T-ViT-14 | $224^2$ | 21.5M | 6.1B | 81.7 |
| | T2T-ViT-19 | $224^2$ | 39.2M | 9.8B | 82.2 |
| | T2T-ViT-24 | $224^2$ | 64.1M | 15.0B | 82.6 |
| | CvT-13 | $224^2$ | 20M | 4.5B | 81.6 |
| | CvT-21 | $224^2$ | 32M | 7.1B | 82.5 |
| | CvT-13 | $384^2$ | 20M | 16.3B | 83.0 |
| | CvT-21 | $384^2$ | 32M | 24.9B | 83.3 |
| Proposed Multi-Stage Conv+TFM | CoAtNet-0 | $224^2$ | 25M | 4.2B | 81.6 |
| | CoAtNet-1 | $224^2$ | 42M | 8.4B | 83.3 |
| | CoAtNet-2 | $224^2$ | 75M | 15.7B | 84.1 |
| | CoAtNet-3 | $224^2$ | 168M | 34.7B | 84.5 |
| | CoAtNet-0 | $384^2$ | 25M | 13.4B | 83.9 |
| | CoAtNet-1 | $384^2$ | 42M | 27.4B | 85.1 |
| | CoAtNet-2 | $384^2$ | 75M | 49.8B | 85.7 |
| | CoAtNet-3 | $384^2$ | 168M | 107.4B | 85.8 |
| | CoAtNet-2 | $512^2$ | 75M | 96.7B | 85.9 |
| | CoAtNet-3 | $512^2$ | 168M | 203.1B | **86.0** |

Table 14: Complete comparison under the ImageNet-21K pre-training + ImageNet-1K finetuning set up. "PT-RA" denotes applying RandAugment during 21K pre-training and "E150" means 150 epochs of pre-training, which is longer than the standard 90 epochs.

| Models | | Eval Size | #Params | #FLOPs | Top-1 Accuracy |
|---|---|---|---|---|---|
| Conv Only | ENetV2-S | $384^2$ | 24M | 8.8B | 85.0 |
| | ENetV2-M | $480^2$ | 55M | 24B | 86.1 |
| | ENetV2-L | $480^2$ | 121M | 53B | 86.8 |
| ViT-Stem TFM Only | ViT-B/16 | $384^2$ | 87M | 55.4B | 84.6 |
| | ViT-L/16 | $384^2$ | 304M | 190.7B | 85.3 |
| Multi-Stage TFM Only | HaloNet-H4 | $384^2$ | 85M | - | 85.6 |
| | HaloNet-H4 | $512^2$ | 85M | - | 85.8 |
| | Swin-B | $384^2$ | 88M | 47.0B | 86.0 |
| | Swin-L | $384^2$ | 197M | 103.9B | 86.4 |
| Multi-Stage Conv+TFM | HaloNet-Conv-H4 | $384^2$ | 87M | - | 85.5 |
| | HaloNet-Conv-H4 | $512^2$ | 87M | - | 85.8 |
| | CvT-13 | $384^2$ | 20M | 16B | 83.3 |
| | CvT-21 | $384^2$ | 32M | 25B | 84.9 |
| | CvT-W24 | $384^2$ | 277M | 193.2B | 87.7 |
| Proposed Multi-Stage Conv+TFM | CoAtNet-2 | $384^2$ | 75M | 49.8B | 87.1 |
| | CoAtNet-3 | $384^2$ | 168M | 107.4B | 87.6 |
| | CoAtNet-4 | $384^2$ | 275M | 189.5B | 87.9 |
| | + PT-RA | $384^2$ | 275M | 189.5B | 88.3 |
| | + PT-RA-E150 | $384^2$ | 275M | 189.5B | 88.4 |
| | CoAtNet-2 | $512^2$ | 75M | 96.7B | 87.3 |
| | CoAtNet-3 | $512^2$ | 168M | 203.1B | 87.9 |
| | CoAtNet-4 | $512^2$ | 275M | 360.9B | 88.1 |
| | + PT-RA | $512^2$ | 275M | 360.9B | 88.4 |
| | + PT-RA-E150 | $512^2$ | 275M | 360.9B | **88.56** |