# OpenReview forum: "CoAtNet: Marrying Convolution and Attention for All Data Sizes"
_NeurIPS.cc/2021/Conference — NeurIPS 2021 Poster_

### Official Review · Reviewer_BJnY · 2021-07-15

**Rating:** 4
**Confidence:** 5

**Summary:**

The paper presents a new architecture combining convolution and transformers block presented as state of the art in terms of FLOPs and accuracy on ImageNet with 3 types of pre-training: ImageNet-1k only, ImageNet-22k and JFT-300M.
In order to explain the logic behind the construction of its architecture the paper tries to answer the following questions:
- How to combine the convolution and self-attention within one basic computational block?
- How to vertically stack different types of computational blocks together to form a complete network?
The paper proposes in the attention modules of its model to associate them with a convolution this is presented as "the key component of the proposed CoAtNet model."



**Limitations And Societal Impact:**

Yes the authors adequately addressed the limitations and potential negative societal impact of their work

**Main Review:**

Strength:
- The idea is interesting
- The results on ImageNet are good.Moreover, the experience seems to be well executed given that the training hyperparameters have been adjusted for each model.

Weakness:
- Some statements in the paper seem to be incorrect:

    1) "CoAtNet achieves 86.0% ImageNet top-1 accuracy without extra data,  and 89.77% with extra JFT data, outperforming prior arts of both convolutional networks and Transformers" L13 . But the paper Meta Pseudo Labels [1] reach 90.2%  ImageNet top-1 accuracy with JFT data, CaiT[2] and NFNets[3] reach 86.5%  ImageNet top-1 accuracy without extra data.  So this assertion of the paper seem incorrect.
    2) "Subsequent works use special regularization and stronger data augmentation to improve the vanilla ViT [16, 17, 18], yet none of these ViT variants could outperform the SOTA convolution-only models on ImageNet classification given the same amount of data and computation [19, 20]" L31 . However the reference 17 ("Going deeper with image transformers" ) gives similar results than reference 20  with similar amount of data and less computation.  So this assertion of the paper seem incorrect.
   3) "When only ImageNet-1K is used for training, CoAtNet achieves 86.0% top-1 accuracy, matching the best public record set by a ConvNet variant NFNet" L57 . NFNet-F5+SAM reach 86.3%  top-1 accuracy and NFNet-F6+SAM reach 86.5%  top-1 accuracy. So this assertion of the paper seem incorrect.

- Significance of the results: There is only ImageNet results there is no evaluation on ImageNet-v2 or ImageNet-real and there is no standard deviation reported. This does not allow to evaluate easily the significance of the results and to know if the method is overfitted.

- Performance comparison: The efficiency of the models is mainly measured in FLOPs and number of parameters but these metrics are far from perfect and it would be interesting to complete them with inference speed or memory consumption at inference time.
Pre-training time Table 5 is measured for few models and only for a pre-training on JFT it would be more informative to have this with more models and on a more widely used dataset.

- Ablation:
1) The paper state "we will use the Transformer block with the pre-normalization relative attention variant in Eqn. (3) as the key component of the proposed CoAtNet model."  L146.  Nevertheless it seems that there is no ablation between this approach and the classical attention. It would be interesting to see how this impacts the performance of the model.

2) Most of the recent paper with vision transformers use the DeiT[5] training procedure. It seems that the parameters used for CoaTNet are different. It would be interesting to know how this impacts performance.

- Related work:  The Convit  paper [4] is quite similar to the CoatNet paper.  Convit have quite similar approach to the relative attention presented in the CoatNet paper. Convit proposes an architecture with blocks closer to convolution at the beginning and transformers block at the end. Moreover, it shows that their approach is more sample efficient than ViT/DeiT. Indeed, they compares it to ViT according to the number of training data which is quite similar to what is done in the CoatNet paper. So the ideas and message of these two papers seem quite similar.
It would be interesting to discuss this paper and compare the CoatNet method with ConViT in Figure 1.


[1] Pham et al., Meta Pseudo Labels

[2] Touvron et al., Going deeper with Image Transformers

[3] Brock et al., High-Performance Normalizer-Free ResNets

[4] Ascoli et al., ConViT: Improving Vision Transformers with Soft Convolutional Inductive Biases

[5] Touvron et al., Training data-efficient image transformers & distillation through attention

**Time Spent Reviewing:**

3

---

> ### Author Response · Authors · 2021-08-10
> **Thanks for the review.**
>
>
> > Some statements w.r.t. SOTA in the paper seem to be incorrect: ...
>
> We meant to express that under a **similar computational budget and training
> setting** (no SAM, no distillation), CoAtNet achieves SOTA-level performance:
>
> -   In the ImageNet-1K setting w/o SAM: CoAtNet-3 and NFNet-5 both got 86.0% with **similar FLOPs / Params**.
> -   When JFT is used, CoAtNet outperformed NFNet & ViT with **similar FLOPs / Params and training budget**.
>
> We will correct this statement to avoid misunderstanding.
>
> --------------------------------------------------------------------------------
>
> > However the reference CaiT [[17]](https://arxiv.org/abs/2103.17239) gives
> > similar results than reference NFNet [[20]](https://arxiv.org/abs/2102.06171)
> > with similar amount of data and less computation. So this assertion of the
> > paper seem incorrect.
>
> -   CaiT [17] requires an **additional distillation step** with a **ConvNet (RegNet)
>     being the teacher** in order to reach the performance level of NFNet [20].
> -   Without the distillation from ConvNet, CaiT [17] is still far from NFNet
>     [20].
>
> --------------------------------------------------------------------------------
>
> > ... it would be interesting to complete them with inference speed or memory
> > consumption at inference time.
>
> We plot the TPU-V3 **inference latency-to-accuracy Pareto curve** (measured with per-core batch
> size = 32) for CoAtNet, NFNet & DeiT (ViT) under ImageNet-1K only setting:
> [Figure link](https://i.ibb.co/23zCKfR/speed.png).
>
> As we can see from the Figure, CoAtNet achieves comparable speed-quality trade-off
> to NFNet, while being significantly better than DeiT (ViT).
>
> --------------------------------------------------------------------------------
>
> > ..., it seems that there is no ablation between this approach (relative
> > attention) and the classical attention.
>
> We do have the ablation of the relative attention in Table 6, where we observe that relative attention does improve the performance.
>
> --------------------------------------------------------------------------------
>
> > Most of the recent paper with vision transformers use the DeiT[5] training
> > procedure. It seems that the parameters used for CoaTNet are different. It would
> > be interesting to know how this impacts performance.
>
> -   We use less augmentation compared to DeiT (we don’t use mixcut, repeated
>     augment), as we hope to simplify the training rather than complicating it.
> -   We also tried to use the reduced set of augmentations to train a DeiT-B
>     baseline, which matches the performance reported (81.8%) in the DeiT-B paper.
>
> --------------------------------------------------------------------------------
>
> > Comparison to the recent ConViT paper.
>
> The performance comparison bwteen ConViT and CoAtNet is summarized
> below:
>
> Models    | Params (M) | FLOPs (B) | Top-1
> --------- | :--------: | :-------: | :---:
> ConViT-S+ | 48         | 10.0      | 82.2
> CoAtNet-1 | 42         | 8.4       | 83.3
> ConViT-B  | 86         | 17.0      | 82.4
> CoAtNet-2 | 75         | 15.7      | 84.1
> ConViT-B+ | 152        | 30.0      | 82.5
> CoAtNet-3 | 167        | 34.7      | 84.5
>
> As we can see, CoAtNet consistently performs better by a large margin given similar FLOPs or Params.

---

> > ### Comment · Reviewer_BJnY · 2021-09-02
> > **Thanks for your response**
> >
> > Thanks for your response, the paper's idea is quite interesting and the imagenet results are good.
> > But the rebuttal does not answer some concern of other reviewers and some of my concern:
> > - Regarding the methodological comparison with the Convit paper, the rebuttal only gave a comparison of the results. This does not answer my question.
> > - Concerning the optimization of the models I am not convinced by the answer of the authors since they adapt their hyperparameters for each network and they use EMA while this is not the case in the DeiT paper.
> >
> > So I keep my initial rating.

---

### Official Review · Reviewer_PojV · 2021-07-16

**Rating:** 5
**Confidence:** 4

**Summary:**

The paper proposes a fusing strategy of convolution (C) and self-attention (SA) layers to get the benefits of both words namely 1. Translation Equivariance (C), 2. Input-adaptive Weighting (SA) and 3. Global Receptive Field (SA). In particular, the authors claimed to improve the generalization and model-capacity of the proposed model through this Conv-Attention module where they have added a weight scalar along with the computed QK^T and then normalized them. Interestingly the authors pointed out this is quite similar to the idea of positional embedding in transformers and referred few NLP papers on that front. Then the authors applied various existing techniques of optimal speed up of attention-based models for better training efficiency and compared among them. The authors did an excellent literature survey and provided thorough results in terms of accuracy performance of the proposed CoAtNet (though I have few concerns and recommendations, see later) on large-scale datasets.

**Ethical Concerns:**

Not much among what is listed. However, I would be happy to see a separate flag for research asking huge compute and thus energy-budget. This is for the conference to keep a close eye on the count of such kind of research. I am not against this line of research, however, I am in favor of keeping a watch on it, so that we can know its impact if it starts affecting on a large scale. I also commit to not judge the paper's contribution with this in mind.

I thus would like to request the NeurIPS PC members to add another ethical concern with the name "environmental impact" (if there is no such track already).

**Ethics Review Area:**

["I don’t know"]

**Limitations And Societal Impact:**

There might be environmental impact, but no social impact to my understanding.

**Main Review:**

> Minor issue, please proofread again to resolve typo issues, for example: "Particularly, VITREL is significantly worse other variants by a large margin"..(L 198). a 'than' is missing here.

> A study of incremental learning to further justify the better model capacity of CoAtNet (or in general attention supported networks) over ConvNets would be interesting.

> The authors said: we also notice that both depthwise convolution and self-attention can be expressed as a weighted sum of values in a pre-defined receptive field (L 107-108).. isn't this valid for any convolution instead? So, from this part of the explanation, it is not clear why the authors shortlist DWC?

> Is the following observation true for self-attentiontion-based vision models with any type of positional embedding? " Due to the usage of absolution positional embeddings, standard Transformer (ViT) lacks this property." (L 125-126), I mean even for attention with specialized pos. embeddings like axial attention [1]?

> The axial attention [1], stand-alone self-attention [2] papers (and there are few others) also talked about and pointed to the issue of the translational equivariance of attention modules and proposed similar pos. embedding layers infused with the Q,K or Q, K, V embedding layers. Thus the finding and the proposed solution does not fall under the category of "novel" approach. The authors deployed the same in Vision transformers which are much larger in param count and thus deploying the idea and validating at such large scale deserves praise, however, I am skeptical about terming the finding as novel and thus concerned about the contribution.

> Please cite [3] in line 155-156.

> The finding of line 178-181 looks quite similar to that of [2].

> I did not understand what might be the reason for Conv+TFM to surpass the performance of COnv only on ImageNet 1K? And if it is so, are the authors hinting at the fact that its better to use conv (or at least no gain in using Conv+TFM) for medium-scale datasets (like ImageNet 1K)?

> Isn't the results of Table 2 is counterintuitive as per the paper's description? If Transformer layers generalize better, then why don't models with higher no of T layers would transfer better (or faster) as well? It would be good it the authors provide explanations to that.

> In another line of research authors of [4] had tried to make the properties of conv learn to SA via a form of distillation. It would be good to see how this solution works as compared to CoAtNet.

[1] Axial-DeepLab: Stand-Alone Axial-Attention for Panoptic Segmentation, ECCV 2020.
[2] Stand-Alone Self Attention for Vision, NeurIPS 2019.
[3] Attention Augmented Convolutional Networks, ICCV 2019.
[4] AttentionLite: Towards Efficient Self-Attention Models for Vision, ICASSP 2021.

**Needs Ethics Review:**

Yes

**Time Spent Reviewing:**

15

---

> ### Author Response · Authors · 2021-08-10
> **Thanks for the review.**
>
>
> > please proofread again to resolve typo issues
>
> Thanks for the catches. We will fix these typos.
>
> --------------------------------------------------------------------------------
>
> > it is not clear why the authors shortlist DWC?
>
> A key difference is whether the weighted sum is performed "per hidden dimension"
> or "cross hidden dimensions".
>
> -   Both Attention and DWC are **per-hidden-dimension** weighted sum.
> -   Standard Convolution is **cross-hidden-dimension** weighted sum.
>
> We will make this point clearer in the updated version.
>
> --------------------------------------------------------------------------------
>
> > Translation equivariance property for attention with specialized pos.
> > embeddings like axial attention [[1]](https://arxiv.org/abs/2003.07853)?
>
> -   If absolute positional embeddings are added to the hidden representation of
>     the Transformer, then it does not enjoy translation equivariance.
>
> -   Similar in spirit to ours, Axial-DeepLab
>     [[1]](https://arxiv.org/abs/2003.07853) utilizes a form of relative
>     attention (what they call position-sensitive attention), which adds
>     positional information to attention logits rather than the hidden states.
>     Hence, the relative attention part enjoys the property translation
>     equivariance.
>
> --------------------------------------------------------------------------------
>
> > I did not understand what might be the reason for Conv+TFM to surpass the
> > performance of COnv only on ImageNet 1K? And if it is so, are the authors
> > hinting at the fact that its better to use conv (or at least no gain in using
> > Conv+TFM) for medium-scale datasets (like ImageNet 1K)?
>
> First of all, the key message we hope to convey is as follows:
>
> -   Fitting Capacity: Conv+TFM > TFM >> Conv
> -   Generalization performance under “limited” data: Conv >= Conv+TFM >> TFM
>
> Given the two inequalities, we would observe the following trend in terms of performance:
>
> -   When the dataset is very small (CIFAR10), we conjecture Conv > Conv+TFM.
> -   The size of ImageNet-1K is probably where Conv and Conv+TFM are getting very
>     close to each other (shown in this work).
> -   When the data size further increases, say ImageNet-21K, we will see Conv+TFM >
>     Conv as the fitting capacity of Conv becomes the bottleneck (shown in this
>     work).
>
> --------------------------------------------------------------------------------
>
> > Isn't the results of Table 2 is counterintuitive as per the paper's
> > description? If Transformer layers generalize better, then why don't models
> > with higher no of T layers would transfer better (or faster) as well?
>
> As presented in the inequality of L197, we actually believe that
> **Convolution**, rather than Transformer, would have better generalization and
> transferibility (i.e. a smaller generalization / transfer gap). Hence, C-C-T-T
> generalizes better than C-T-T-T.

---

> > ### Comment · Reviewer_PojV · 2021-09-02
> > **Okay but not good enough**
> >
> > Dear authors,
> >
> > I appreciate your effort in clarifying few of my concerns. However, my major concern about the contribution of this work still remains. The contribution of pos embedding is not at all novel even in vision (the authors have also rightfully agreed to this in one of their rebuttal comments " *Similar in spirit to ours, Axial-DeepLab [1] utilizes a form of relative attention..*"). Earlier other reviewers have already pointed similar embedding to contribute better in NLP as well.  Also, few earlier works in ViT research has hinted at the generalization capacity aspect of transformer and vision based models. The results are good though I understand is not the SOTA, however I am not in favor of judging the contribution solely in terms of results being SOTA or not. Overall, I find no reason to promote this paper above acceptance threshold as to me this paper still remains as a paper of limited contribution.

---

### Official Review · Reviewer_5AC9 · 2021-07-16

**Rating:** 6
**Confidence:** 4

**Summary:**

In this paper, the authors provide extensive empirical study results on how to combine transformer blocks and convs for best performance of image recognition. The authors mainly search for 4 different configurations with different numbers of conv-stages and numbers of transformer stages. The conv stages always come before transformer stages. The authors finally present a series of models based on the search results. The final models outperforms existing ViT based or conv-based models on multiple dataset-size and resolution settings.

**Limitations And Societal Impact:**

I think the main limitation is the vague description of the exact model specification. Regarding societal impact, the research seems computationally quite expensive to conduct (thus potentially consuming much energy). Other than that, I don't see this paper to be particular concerning compared to other deep network design papers.

**Main Review:**

**[originality: low]**
 - This paper does not propose a new method or architecture building block.
 - The training techniques, search method, and evaluation methods are all standard.
 - Combining conv and attention is also not new (e.g. non-local neural network (Wang et al., CVPR 2018) does this).
 - The method presented in 3.1 is effectively a relative positional encoding similar to what's proposed in [37].

Thus overall, I don't see this paper to be particularly original or novel.

**[quality: medium - high]**

On the other hand, the empirical study presented in this paper can be quite useful. The choice of ablation study and analysis gives useful insight on ViT+conv-based vision model design. The results convincingly demonstrate the strength of the proposed method. I also find the ablation study or analysis (e.g. Table 1) useful.

**[clarity: medium]**
 - While in Table 3, a summary of the proposed models is provided, more details about the architecture specifications are required. At the current level of details, reproducing any results would be difficult.
 - The writing is overall clear and easy to read.
 - While 3.1 is interesting to read, the similarity to relative positional encoding is vague. I think a more straightforward and concise description and a side by side comparison to commonly-used positional encoding would make the method easier to understand.

**[significance: high]**

The area of ViT-related model design is promising for computer vision research and introduces many interesting open questions. This paper provides useful empirical study that can serve as useful reference for future research.



**Time Spent Reviewing:**

1.5 hours

---

> ### Author Response · Authors · 2021-08-10
> **Thanks for the review.**
>
>
> > I think the main limitation is the vague description of the exact model
> > specification.
>
> -   A more detailed description of each module is presented in Appendix A.1. We
>     hope the details there could help.
> -   In addition, we will try to improve the writing in the main text to make the
>     model specification clearer.

---

> > ### Comment · Reviewer_5AC9 · 2021-08-31
> > **Re: Thanks for the review**
> >
> > Thanks for sharing the author feedback.  Yes, Appendix A1 and improving the writing in the main paper would be helpful.
> >
> > After reading all reviews and all author feedback, I still think the empirical results presented in this paper will be useful for future research on vision architecture design. I thus keep my recommendation unchanged.

---

### Official Review · Reviewer_qHHe · 2021-07-19

**Rating:** 5
**Confidence:** 4

**Summary:**

This paper proposes to marry convolution with self-attention in order to combine the strengths from both architectures (good inductive bias, larger model capacity). By vertically stacking depth-wise convolution layers and self-attention layers, the resulting CoAtNet gets better generalization ability, capacity, and efficiency. The author carefully designs and tests the layout choices and finally chooses C-C-T-T as the architecture of CoAtNet. From the experimental results, the proposed CoAtNet achieves 86.0% ImageNet top-1 accuracy with 167M parameters and 512 as input size and outperforms previous Convolution and Vision-Transformer baselines. Further extensive experiments and ablation studies show that CoAtNet enjoys good generalization ability like Convolution Networks and superior model capacity as Transformers, and possesses better efficiency over Transformers.

**Ethics Review Area:**

["I don’t know"]

**Main Review:**

Strengths:
1) The paper is generally well-written and easy to follow.
2) This paper works on an interesting problem—combing Convolution and Self-attention to get the strengths from both operations. The resulting CoAtNet effectively solves the drawbacks of Vision Transformers (need more data and larger parameters to train) and enjoys good generalization ability, model capacity, and efficiency.
3) Extensive experiments are conducted to confirm the good CoAtNet layout and architecture design, the effectiveness and efficiency of CoAtNet over previous Convolution-based and Transformer-based approaches.
4) The proposed CoAtNet reached a high performance of 86.0% ImageNet top-1 accuracy without extra data and compares to NFNet-F5, CoAtNet requires less than a half number of parameters and is faster. If trained with ImageNet 21K data and JFT data, CoAtNet could reach 88.56% and 89.77% top-1 accuracy respectively, and demonstrate huge data efficiency compared to ViT.

Weaknesses and Questions:
1) CoAtNet only explores ImageNet classfication. Compared with the papers (such as PVT/Swin/BoT) which perform extensive experiments on detection, segmentation, self-supervised learning, CoAtnet is very weak in this perspective.
2) The main idea of unifying convolution with transformers has been explored in the following NLP papers. The author needs to discuss the relationship between them. In my opinion, the core idea of CoAtNet is the same as proposed in NLP research. I am prone to weakly reject this paper as lack of novelty and lack of discussion with paper in NLP:
[1] Zhao, Guangxiang, Xu Sun, Jingjing Xu, Zhiyuan Zhang, and Liangchen Luo. "Muse: Parallel multi-scale attention for sequence to sequence learning." arXiv preprint arXiv:1911.09483 (2019).
[2] Wu, Felix, Angela Fan, Alexei Baevski, Yann N. Dauphin, and Michael Auli. "Pay less attention with lightweight and dynamic convolutions." arXiv preprint arXiv:1901.10430 (2019).
[3] Convolutions and Self-Attention: Re-interpreting Relative Positions in Pre-trained Language Models. Tyler A. Chang, Yifan Xu, Weijian Xu and Zhuowen Tu. ACL-IJCNLP 2021.
3) A minor concern: The author said that "CoAtNets achieve state-of-the-art performance under different resource constraints across various datasets." However, the author forgot to compare with the following ViL paper. When we set 224 as the input size, then compare ViL-S (Top-1 Acc: 82.0, #Params: 24.6M, #FLOPs: 4.9B) and CoAtNet-0 (Top-1 Acc: 81.6,  #Params: 25M, #FLOPs: 4.2B). CoAtNet-0 seems not to outperform ViL-S.
[4] Pengchuan Zhang, Xiyang Dai, Jianwei Yang, Bin Xiao, Lu Yuan, Lei Zhang, and Jianfeng Gao. Multi-scale vision longformer: A new vision transformer for high-resolution image encoding. arXiv preprint arXiv:2103.15358, 2021
4) The code is not provided with the supplementary materials, but the checklist said yes.
5) Why pre-normalization is better than post-normalization is not very clear to me, could you elaborate more on this?


**Time Spent Reviewing:**

3 hours

---

> ### Author Response · Authors · 2021-08-10
> **Thanks for the review.**
>
>
> > CoAtNet only explores ImageNet classfication. Compared with the papers (such
> > as PVT/Swin/BoT) which perform extensive experiments on detection,
> > segmentation, self-supervised learning, CoAtnet is very weak in this
> > perspective.
>
> The reason we focus on image classification is that it offers us the cleanest
> setup to systematically study the interaction between convolution and
> self-attention layers. This narrower focus also enables us to conduct in-depth
> analysis related to capacity, scalability and transferability of the models. The
> very same paradigm has been successful in several impactful papers such as
> EfficientNets and ViT.
>
> --------------------------------------------------------------------------------
>
> > The main idea of unifying convolution with transformers has been explored in
> > the following NLP papers
> > [[1]](https://arxiv.org/abs/1911.09483)[[2]](https://arxiv.org/abs/1901.10430)[[3]](https://arxiv.org/abs/2106.05505).
>
> -   First of all, a core contribution of CoAtNet is studying how to properly
>     stack convolution and transformer blocks into a complete network to achieve
>     good capacity as well as generalization, not just the integration of
>     convolution and attention.
> -   In addition, we do NOT try to claim relative attention is our invention.
>     Instead, we hope to point out our observation that while there are many
>     efforts trying to put convolution priors into the attention, the well
>     established relative attention is a “natural” combination with great
>     performance.
>
> --------------------------------------------------------------------------------
>
> > ... However, the author forgot to compare with the following ViL paper
>
> Thanks for the pointer. As there are too many new ViT variants recently, we may
> have missed some of them.
>
> --------------------------------------------------------------------------------
>
> > The code is not provided with the supplementary materials, but the checklist
> > said yes
>
> That must be a typo. We will correct that statement, and we will open
> source the code.
>
> --------------------------------------------------------------------------------
>
> > Why pre-normalization is better than post-normalization is not very clear to
> > me, could you elaborate more on this?
>
> There is no rigorous proof on this point. Our rough logic is as follows.
>
> -   For the Transformer block, x + F(LN(x)) is more popular than the original
>     LN(x + F(x)) recently. The reasoning follows the same intuition of the
>     [pre-activation](https://arxiv.org/pdf/1603.05027.pdf) formulation in
>     ResNet.
> -   As for the MBConv block, we conjecture that making it homogenous with the
>     pre-norm Transformer block could potentially prevent over-optimizing some
>     part of the model, and hence lead to more stable optimization.

---

### Review · Ethics_Reviewer_kHQK · 2021-08-09

**Recommendation:**

A discussion of broader impacts and potential mitigation strategies at the end of the manuscript would be sufficient

**Ethical Issues:**

Yes

**Ethics Review:**

I agree with the assessments of the technical reviewers that the ethical considerations relevant to this work are the environmental and financial costs associated with developing and deploying large models. These issues are not discussed in the manuscript.

---

### Review · Ethics_Reviewer_CDak · 2021-08-10

**Recommendation:** No need to address them.

**Ethics Review:**

The one reviewer has suggested it requires Ethical Review because it is a large language model, and therefore has potential harmful effects on the environment. I do not think this energy-consumption concern fits within any of the negative impacts listed in the ethical guidelines for the conference.

---

### Decision · Program_Chairs · 2021-09-28

**Decision:**

Accept (Poster)

**Comment:**

This paper has received slightly negative ratings - the main concern being the lack of novelty, as well as several ambiguous claims. Following the rebuttal and discussion phase, the reviewers acknowledged the quality of the ImageNet results, but some questions remained unanswered. For this reason, I recommend rejection, and encourage the authors to take the reviewer's comments into account to improve the manuscript and resubmit to another venue.

**Consistency Experiment:**

NeurIPS has a long history of experimentation. In 2014, NeurIPS ran an experiment in which 10% of submissions were reviewed by two independent committees to quantify the randomness in the review process. This year, we repeated a variant of this experiment to see how the quality of the review process has changed over time.  This paper was part of the experiment and was therefore assigned to two committees (consisting of reviewers, an Area Chair, and a Senior Area Chair) that reached independent decisions.  If both committees made the same recommendation, this recommendation was followed. If a single committee recommended acceptance, the paper was accepted (with the exception of a few cases in which the other committee identified what we considered a fatal flaw, e.g., an error in a key result).

This copy’s committee reached the following decision: **Reject**

The other committee assigned to the paper recommended **Accept (Poster)**.  You can find the other set of reviews, along with any follow up discussion with the authors here:
https://openreview.net/forum?id=dUk5Foj5CLf